# Markers of Genetic Variation in Blue Gourami (*Trichogaster trichopterus*) as a Model for Labyrinth Fish

**DOI:** 10.3390/biology10030228

**Published:** 2021-03-16

**Authors:** Gad Degani, Isana Veksler-Lublinsky, Ari Meerson

**Affiliations:** 1MIGAL–Galilee Research Institute, P.O.B. 831, Kiryat Shmona 1101602, Israel; gad@migal.org.il; 2Faculty of Sciences, Tel-Hai Academic College, Upper Galilee 1220800, Israel; 3Department of Software and Information Systems Engineering, Ben Gurion University of the Negev, P.O.B. 653, Beer-Sheva 8410501, Israel; vaksler@post.bgu.ac.il

**Keywords:** mitochondrial genes, cytochrome b, 12S rRNA gene, Anabantoidei, growth hormone, reproduction

## Abstract

**Simple Summary:**

This review is a summary of recent studies of genes, many of them involved in growth and reproduction, which can be used for distinguishing between species of the Anabantoidei suborder of fish, focusing on the Blue Gourami as a model species. This is important in both basic science and aquaculture applications.

**Abstract:**

Markers of genetic variation between species are important for both applied and basic research. Here, various genes of the blue gourami (*Trichogaster trichopterus*, suborder Anabantoidei, a model labyrinth fish), many of them involved in growth and reproduction, are reviewed as markers of genetic variation. The genes encoding the following hormones are described: kisspeptins 1 and 2, gonadotropin-releasing hormones 1, 2, and 3, growth hormone, somatolactin, prolactin, follicle- stimulating hormone and luteinizing hormone, as well as mitochondrial genes encoding cytochrome b and 12S rRNA. Genetic markers in blue gourami, representing the suborder Anabantoidei, differ from those in other bony fishes. The sequence of the mitochondrial cytochrome c oxidase subunit 1 (COI) gene of blue gourami is often used to study the Anabantoidei suborder. Among the genes involved in controlling growth and reproduction, the most suitable genetic markers for distinguishing between species of the Anabantoidei have functions in the hypothalamic–pituitary–somatotropic axis: pituitary adenylate cyclase-activating polypeptide and growth hormone, and the 12S rRNA gene.

## 1. Introduction

Genetic variability between the organisms refers to sequence differences between their genomes, part of which is reflected in the sequence of protein-coding genes. This variation in DNA sequence can be used as a marker to distinguish between organisms, including fishes, at all systemic levels [1,2,3,4,5,6,7,8]. Compared to the much-studied genetic markers in some fish species of economic value, little research has been done with the group of labyrinth fishes. Blue gourami (*Trichogaster trichopterus*) is a labyrinth fish of the suborder Anabantoidei, order Perciformes. It is a small tropical freshwater fish characterized by the presence of a chamber, or labyrinth, above the gills for the uptake of oxygen from the air for breathing. The labyrinth enables adaptation to life in water with low dissolved oxygen by partial air breathing. Anabantoid fishes are geographically distributed in central Africa, India, and southern Asia [9] (Figure 1). In their natural habitat, they adapt to an unpredictable environment in which water-dissolved oxygen concentration varies throughout the year and can reach very low concentration [10]. The 16 known genera of anabantoid fishes contain about 80 species (FishBase, Nelson et al. Fishes of the World [10]). However, the systematic characters of the labyrinth fishes have not been agreed upon and many synonyms are used. According to Vierke [10], taxonomists classify the labyrinth fishes into four families: Anabantidae (genera *Sandelia*, *Ctenopoma*, *Anabans*), Belontiidae (genera *Trichopsis*, *Trichogaster*, *Sphaerichthys*, *Pseudosphromenus*, *Parosphromenus*, *Malpulutta*, *Helostoma*, *Ctenops*, *Colisa*, *Betta*, *Belonti*a), Osphronemidae (genus *Osphronemus*), and Helostomatidae (genus *Helostoma*).

Labyrinth fishes have two different stages in their life cycle: (i) before labyrinth organ development, from eggs to juveniles, when oxygen is absorbed over the entire surface by diffusion; and (ii) after labyrinth development, when the organ becomes important for breathing [10]. The adaptation to the development of eggs and fry in water with low oxygen concentration involves laying eggs in a bubble nest which supplies oxygen to the eggs and larvae [10]. In natural habitats, when there is a low density of mature males, they become territorial by building a bubble nest and protecting it from other males (Figure 2) [12,13].

After courting and fertilization, the female swims under the bubble nest and spawns eggs into it. The male guards the nest with the eggs. If an egg falls out, the male returns it to the nest. The male also protects the young fish, immediately after hatching (Figure 3).

Ornamental fish populations in their native habitats, mainly in tropical areas, have declined due to overfishing for their sale on the tropical fish market. Fish of the suborder Anabantoidei are important in the ornamental fish industry and have long been produced in aquaculture [10]. The introduction of *T. trichopterus* in Florida is considered to have failed, according to US Fish and Wildlife Service, June 2019 [11].

## 2. Sequencing Analysis of the 12S rRNA and Cytochrome b Gene Variations in Blue Gourami

Information on the molecular variation of species belonging to the Anabantoidei in the order Perciformes (perch-like-fishes) is very limited [2,15,16]. Polymorphisms in several enzymes can be utilized as a genetic marker for these Anabantoidei species belonging to the Perciformes [5,17].

Thus, two Anabantoidei species—*Trichogaster trichopterus* and *Trichogaster leerii*—are similar in morphology and in their geographical distribution. *Colisa lalia* does not exhibit an overlapping geographical distribution, but shows a high similarity to *Betta betta*, which is distributed in the small area covered by the genus *Trichogaster*. The gene loci of species belonging to the genus *Colisa* showed a higher degree of similarity to those of the *Trichogaster* species than to those of the *Betta* species [5]. Population structures of *Trichogaster pectoralis* collected from five locations in Thailand [17] were also studied by isozyme analysis. The highest genetic identity coefficient was found between Samutprakan and Pitsanulok populations in Thailand; the lowest between Pitsanulok and Pattanee populations.

Several sets of degenerate oligonucleotides were used by Degani [2] to clone DNA fragments encoding portions of the cytochrome b and *12S rRNA* genes. These genes were used to examine the genetic variation between species of the Anabantoidei by mitochondrial gene-sequencing analysis. Results demonstrated a similarity between the gene sequences of the various Belontiidae species, leading to the finding that these genes could serve as molecular markers for the systematic study of Belontiidae reproduction. The cytochrome b sequences of Anabantoidei fish examined in that study are shown in Figure 4.

Based on this gene, the most similar species were *Trichogaster trichopterus* (gold) and *Trichogaster trichopterus* (blue) (100%). *Trichogaster leerii* was less similar to these (86.0%), and even lower similarity was found between the species *T. trichopterus* and *Trichogaster labiosus* (85.6%). The least similarity was observed between *Betta betta* and the genera *Colisa* (50.2%) and *Trichogaster* (60.1%). The similarity value for the cytochrome b gene of *Macropodus opercularis* was between that of *Betta* and *Colisa* [2]. The phylogenetic tree for *cytochrome b* is shown in Figure 5 [2].

Additionally, sequences of the rRNA *12S* gene from *T. trichopterus* (gold), *T*. *trichopterus* (blue), *T*. *leerii*, *C*. *lalia*, and *B. betta* are shown in Figure 6, and the nucleotide-similarity phylogenetic tree is presented in Figure 7 [2]. The phylogenetic tree results for the 12S rRNA gene were very similar to those for *cytochrome b. 12S rRNA gene* sequence similarity between *Trichogaster* species was high (91.4%–100%), and there was less similarity between this genus and C. lalia (88.4%). In addition, there was less similarity between *T. trichopterus* and *C. lalia* than between *T. trichopterus* and *B. betta* (82.6% and 84.0%, respectively) [2].

The cytochrome b and *12S rRNA gene* sequences of the Anabantoid fishes were compared to those of other fishes and presented as a phylogenetic trees of nucleotide similarity (Figure 8 and Figure 9). In this comparison, some fishes that do not belong to the Anabantoidei seemed to have high similarity to the Anabantoid fishes.

## 3. Hypothalamus–Pituitary-Gonad (HPG) Axis Genes as Molecular Markers in Blue Gourami and Other Anabantoid Species

The genes involved in the HPG are important in the breeding of fishes and therefore their sequences have been studied. In teleosts, as in other vertebrates, kisspeptin (Kiss) has recently received considerable attention as a potential key player in the indirect neuroendocrine control of reproduction [18]. Kiss is a member of the RFamide peptide family. Originally identified as a metastasis suppressor in mammals, the *Kiss1* gene produces several Kiss peptides in mammals. Kiss54 and its endogenous variants, Kiss14, Kiss13, and Kiss10, are generated by proteolytic cleavage of the Kiss precursor derived from the *Kiss1* gene. They share a common core 10-amino acid (aa) sequence (Kiss10) at their C-terminal end, which allows them to bind to their cognate G-protein-coupled receptor (GPR54) or Kiss receptor (Kiss1r) [19]. Thus, Kiss1 controls the HPG axis, acts on the caudal hypothalamus and seems to affect receptors of gonadotropin-releasing hormone (GnRH) [20]. It controls the release of the pituitary gonadotropins follicle-stimulating hormone (FSH) and luteinizing hormone (LH), which in turn control gametogenesis [21] through steroids [21,22,23,24]. Studies on Kiss in teleosts have shown variation in their involvement in reproduction; however, more detailed studies are required due to the relatively large size of this class and the inherently high variation in hormones involved in reproduction. The brain of zebrafish (*Danio rerio*), one of the most intensively studied model fish, has two Kiss genes, *Kiss1* and *Kiss2*, and two Kiss receptors (GPR54): Kiss1r and Kiss2r [19,25]; this is similar to other fishes such as lamprey (*Petromyzon marinus*) [25], medaka (*Oryzias latipes*) [26], goldfish (*Carassius auratus*) [27] and striped bass (*Morone saxatilis*) [28]. We studied the DNA sequence of the brain *Kiss2* and the two Kiss receptors (GPR54), *Kiss1r* and *Kiss2r*, in blue gourami [29,30]. The obtained partial sequences of *Kiss2* were compared with homologous sequences from a number of other fish species (Figure 10).

There was a low degree of similarity for both nucleotide and amino acid sequences of Kiss2 between the blue gourami and other fish species (Figure 10). A higher degree of similarity was found between the Kiss2r of blue gourami and those of the other fish species (Figure 11). The results show that *Trichogster trichopterus* Kiss2r’s amino acids sequence is very different from those of the species *Tetraodon nigroviridis*, *Xenopus tropicalis*, *Scomber japonicus*, and *Gasterosteus aculeatus* and closer to the species *Danio rerio*, *Carassius auratus*, and *Dicentrarchus labrax*.

The obtained sequences of Kiss1r in the blue gourami were also compared with homologous sequences in a number of other fish species. There was a low degree of similarity for both nucleotide and amino acid sequences between the blue gourami sequence and those of other fish species (Figure 12).

GnRH plays a central role in the control of vertebrate reproduction by affecting various other hormones, e.g., gonadotropins and the growth hormone (GH) family, which in turn regulate gametogenesis and steroidogenesis [31,32]. cDNA cloning of three GnRH forms (GnRH1, GnRH2, and GnRH3) of blue gourami was described by Levy et al. [31,32,33] (Figure 13 and Figure 14).

As shown in Table 1, the degree of identity between the blue gourami GnRH3 preprohormone amino acid sequence and that of other fishes was 80–82.2% [31]. There was a higher degree of identity at both the nucleotide and amino acid levels between the blue gourami sequence and the respective sequences of *Dicentrarchus labrax* and *Pagrus major* (Order Perciformes). In other perciforms (*Cynoscion nebulosus*, *Rachycentron canadum*, *Sciaenops ocellatus*, and *Mugil cephalus*), a higher degree of amino acid similarity was found, as compared to the more variable nucleotide sequence [31,32,33].

## 4. DNA Sequences of FSH and LH as Molecular for Genetic Similarity between Blue Gourami and Other Fish Species

FSH and LH are gonadotropins that control gametogenesis in fishes. The sequences of those genes are described in many fishes. FSH and LH cDNAs from the pituitary gland of blue gourami (*T. trichopterus*), encoding the α and β subunits of these hormones, were cloned [34]. The two cDNAs were sequenced and analyzed. The deduced amino acid sequences of both FSH and LH cDNAs are presented in Figure 15.

Comparison of FSH-β and LH-β are shown in Figure 16. The blue gourami FSH-β was most similar to its striped bass counterpart, with the two polypeptides sharing 73% of their residues. The lowest similarity was found between blue gourami and of goldfish FSH-β, with only 44% similarity (Figure 17A) [34]. A dendrogram, graphically representing these polypeptide comparisons, showed highest similarity between blue gourami and striped bass FSH-β, with 84% identical residues (Figure 17A). For LH-β, the lowest similarity was found with the *Coregonus migratorius*, a whitefish of the Family Salmonidae. (only 65% identical residues). The dendrogram showing the relationships between the LH-β polypeptides is presented in Figure 17B [34].

## 5. Hypothalamic–Pituitary–Somatotropic (HPS) Axis Genes as Markers for Genetic Variation between Blue Gourami and Other Fish Species

Pituitary adenylyl cyclase-activating polypeptide (PACAP) and PACAP-related peptide (PRP) are members of the secretin/glucagon/vasoactive intestinal polypeptide family. PACAP was first isolated from the ovine hypothalamus [36], and its sequence has also been determined in representative species of non-mammalian vertebrates. In fishes, PRP (formerly known as GHRH or GHRH-like peptide) was first isolated from the hypothalamus of common carp [37]. In teleosts, the complete sequence of the PRP–PACAP cDNA has been cloned from several species [36]. In all vertebrates, the PACAP preprohormone consists of a signal peptide, a cryptic peptide, a PRP segment and a PACAP segment. Two processing sites, Lys–Arg and Gly–Arg–Arg, between the PRP and PACAP fragments and in the PACAP sequence, respectively, result in three peptides: PRP, PACAP27 and PACAP38. In several teleosts, there are two transcripts of PRP–PACAP as a result of exon skipping. This stems from alternative splicing resulting in the partial excision of exon 4, which encodes part of PRP (residues 1–32), leaving the PACAP-encoding region in the correct reading frame [38]. PACAP can bind specifically to three G-protein-coupled receptor (GPCR) variants (PACAP-Rs) respectively termed: PAC1-R, VPAC1-R, and VPAC2-R [36]. Differential distribution of PACAP-R has been identified in the brain, pituitary, heart, spleen, liver, gut, gills, kidney, skin, blood, and gonads [36]. PRP-R also belongs to the GPCR family, but has only been identified in non-mammalian species, e.g., in the goldfish pituitary [39]. These findings suggest that evolutionary pressure has acted to preserve the bioactive sequence of PACAP, indicating that the peptide must exert important physiological functions. PACAP and PRP are involved in growth promotion and GH control. In teleosts, PACAP is involved in various physiological processes, such as brain development, ventilation and cardiac baroreflex control, digestive physiology, immune response, food intake, and growth promotion [40].

The structures of the PRP–PACAP cDNA are presented in Figure 18A [36], and the full-length cDNA sequence of the PRP–PACAP precursor, compiled from data obtained from 5’ and 3’ RACE, and its deduced amino acid sequence, are shown in Figure 18B [36]. The nucleotide and amino acid sequences of this gene in blue gourami were compared with homologous sequences from a number of other fish species (Figure 19) [36]. Sequence alignment of the PRP amino acid sequence revealed that only the first 32 aa at the N terminus are highly conserved in closely related fishes. There was 80.0–93.3% sequence identity between the blue gourami PRP sequence and that of other teleosts, whereas there was only 31.6–56.6% sequence identity between other tetrapods and fishes (Figure 19A). On the other hand, the amino acid sequence of PACAP is highly conserved in mammals and fishes.

## 6. GH and Prolactin (PRL) Family Hormones as Genetic Variation Markers for Blue Gourami and Other Anabantoid Fishes

Growth hormone (GH) is a single-chain polypeptide. Together with prolactin, growth hormone and somatolactin, it forms a family of related polypeptide hormones whose sequences seem to have evolved from a common ancestor [31,33,36]. GH has been studied extensively, and cDNA sequences are available for blue gourami (Figure 20) and many other teleosts. The study of GH expression in relation to growth and the reproductive cycle could contribute to an understanding of the interactions between somatotropic and gonadotropic axes at the pituitary level in order to elucidate the effects of GH on fish reproduction [31,33,36]. Other HPS axis genes that control growth and that are involved in cell division have been sequenced in blue gourami but less in other species of Anabantoid fishes [7,31,36,41,42].

In a comparison of the deduced amino acid sequence of bgGH to other fish sequences found in the GenBank database, bgGH was most similar to that of *Sparus aurata* (86% identical residues) and least similar to *Anguilla japonica* (43% identity) (Figure 21). The sequence similarity was in accordance with prevailing systematics [41].

Prolactin is a hormone involved in a large number of biological processes, including growth, reproduction and osmoregulation in fish. The complete cDNA of the blue gourami PRL (bgPRL) was cloned by RACE PCR [43]. The deduced amino acid sequence of bgPRL was compared with that of homologous subunits from a number of other fish species (Table 2). The highest degree of homology was between the gourami and the Perciforms *Perca flavescens*, *Dicentrarchus labrax*, and *Sparus aurata*, followed by members of the order Salmoniformes (66%), Siluriformes, and Cypriniformes (Table 2) [43]. The lowest level of homology was observed with the Anguilliformes (Table 2) [43].

Somatolactin (SL), a specific pituitary hormone belonging to the PRL superfamily, is involved in environmental adaptation, osmoregulation, reproduction, and fatty acid metabolism. The cDNA of SL from blue gourami was cloned and subjected to DNA sequence analysis [44,45]. The partial cDNA sequence of SL was compared to those of other fish species (Figure 22).

## 7. Mitochondrial Cytochrome c Oxidase Subunit 1 (COI) Gene as a Variation Marker for Blue Gourami and Other Anabantoid Fishes

COI is another genetic marker widely used to identify the kinship of fish species (Figure 24 and Figure 25).

## 8. Discussion

This review presents sequence analysis of genes involved in various growth and reproductive processes that may serve as markers of genetic variation between labyrinth fishes of the suborder Anabantoidei and other fish species and groups (Summary, Table 3). The various molecular markers in blue gourami are mitochondrial or nuclear DNA sequences, but not microsatellites or single-nucleotide polymorphisms. In the present review, most of the genetic markers were associated with growth (PACAP, GH and PRL) and reproduction (GnRH1, GnRH2, GnRH3, LH, and FSH). Only two mitochondrial genes (*cytochrome b* and *12S*) were studied. Isozyme markers also have been studied in labyrinth fishes [5]. Thus, one can compare genomic markers with mitochondrial ones. The similarity of *cytochrome b* sequence between labyrinth and other fish species was between 86% and 46%, and of the 12S rRNA gene, between 91% and 40%. Sequence similarities between the blue gourami and other fish species for the hormone-encoding genes seemed to be lower than for the mitochondrial genes (Table 3). Among the genes involved in controlling growth and reproduction, those with the highest sequence similarity between species were those from the HPS axis. The findings based on DNA sequence comparison presented in this review are in agreement with many other studies [1,2,3,4,5,6,7,8,48,49]. All of the tested genes in blue gourami had high similarity to their counterparts in other fishes in the order Perciformes, to which the blue gourami belongs [10], and some of them can be useful as genetic markers in other classes of fish.

For example, the cytochrome b sequence was used to study genetic variation of the genus *Capoeta* and specifically the species *C. damascina* [49,50]. *C. damascina* is one of the most common freshwater fish species found in a wide range of isolated water bodies throughout the Levant, Mesopotamia, Turkey, and Iran. Prior to these studies, *C. damascina* was not a well-defined species. Likewise, the COI gene is widely used in assessing genetic variation in both geographical distribution and aquaculture contexts [51,52]. This gene is suitable for the separation of different species belonging to the genus Trichogaster, but not of the order Anabantiformes (Figure 23 and Figure 24).

## Figures and Tables

**Figure 1 biology-10-00228-f001:**
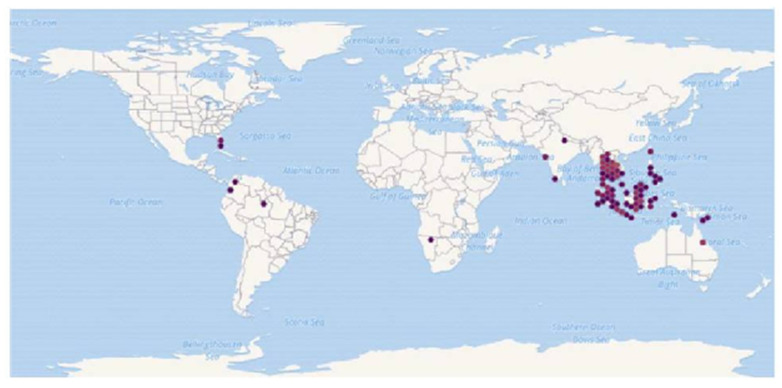
Known global distribution of *Trichogaster trichopterus*. Locations are in Australia, Papua- New Guinea, Indonesia, Malaysia, the Philippines, Taiwan, Vietnam, Cambodia, Laos, Thailand, Myanmar, India, Namibia, Colombia, Brazil, and the United States. Map from the GBIF Secretariat [11].

**Figure 2 biology-10-00228-f002:**
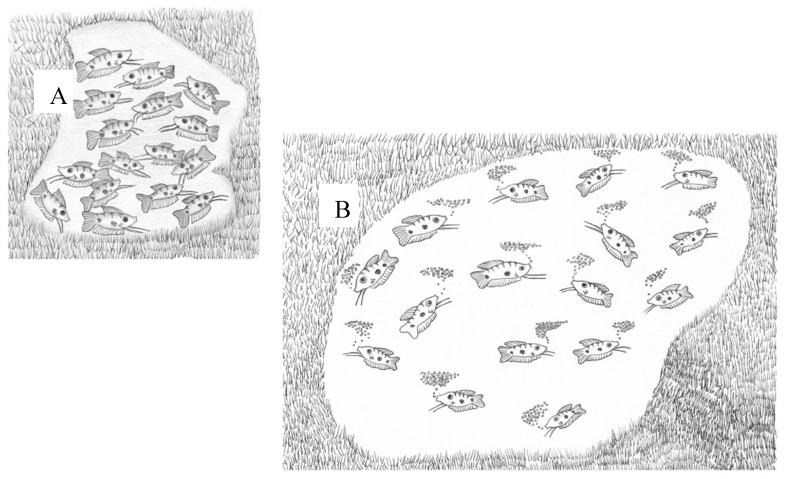
A scheme of two habitats, with high (**A**) and low (**B**) densities of males. At high density, the male does not build a nest. At low density, the male builds a nest and sexual behavior is initiated [10,13,14].

**Figure 3 biology-10-00228-f003:**
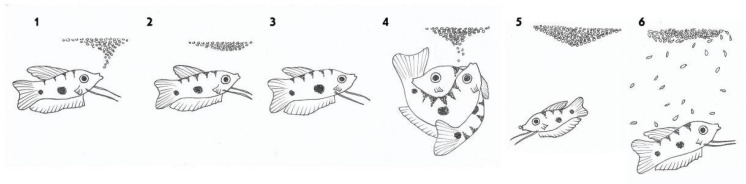
Sexual behavior of male blue gourami during the reproductive cycle. (1) The male builds a nest. (2) and (3) The male courts the female under the nest. (4) Fertilization takes place and the fertilized eggs float up and stick to the bubble nest. (5) The male guards the eggs in the nest. (6) The male further protects the fry immediately after hatching while they swim in the nest area [10,13].

**Figure 4 biology-10-00228-f004:**
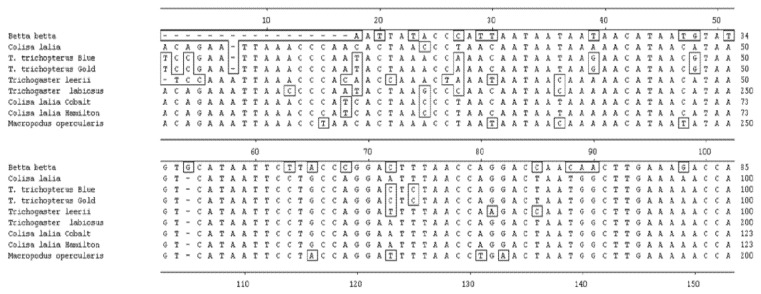
Comparison of cytochrome b nucleotide sequences (150 bp) in *T. trichopterus* (blue and gold gourami), *T. leerii* (pearl gourami), *C*. *lalia* (dwarf gourami), *T. labiosus* (*Colisa labiosa*), *B. betta* (fighting fish), and *M. opercularis* [2].

**Figure 5 biology-10-00228-f005:**
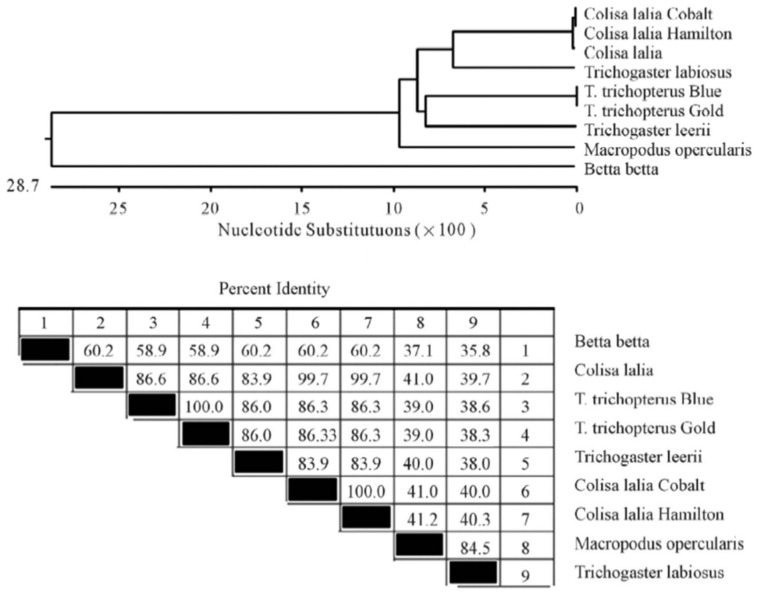
Composite phylogenetic tree constructed from the analysis of cytochrome b sequences of six species: *T. trichopterus* (blue gourami and gold gourami), *T. leerii* (pearl gourami), *C. lalia* (dwarf gourami), *T. labiosus* (*C. labiosa*), *B. betta* (fighting fish), and *M. opercularis*. The phylogenetic tree was constructed by Clustal W and analysis alignment methods in the MegAlign program (DNASTAR) [2]. Branch lengths represent evolutionary distances. Percent DNA sequence identities are also shown [2].

**Figure 6 biology-10-00228-f006:**
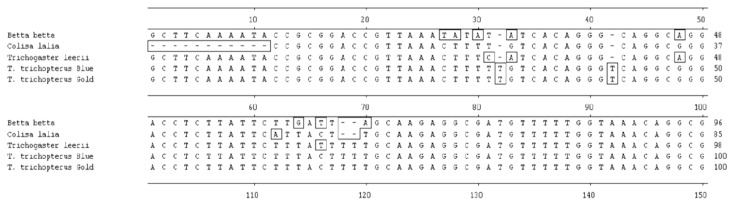
Comparison of the *12S rRNA gene* nucleotide sequences (150 bp) in *Trichogaster trichopterus* (blue gourami and gold gourami), *T. leerii* (pearl gourami), *C. lalia* (dwarf gourami), and *B. betta* (fighting fish) [2].

**Figure 7 biology-10-00228-f007:**
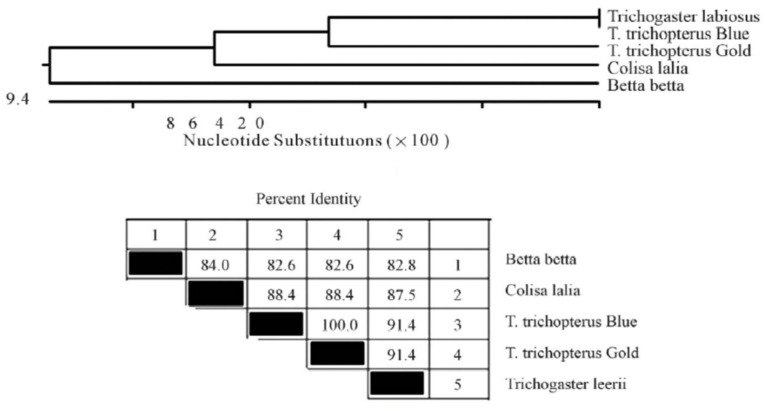
Phylogenetic tree constructed from analysis of *12S rRNA* gene sequences of *T*. *trichopterus* (blue gourami and gold gourami), *T. leerii* (pearl gourami), *C. lalia* (dwarf gourami), and *B. betta* (fighting fish). The phylogenetic tree was constructed by ClustalW and analysis alignment methods in the MegAlign program (DNASTAR). Branch lengths represent evolutionary distances. Percent DNA sequence identities are also shown [2].

**Figure 8 biology-10-00228-f008:**
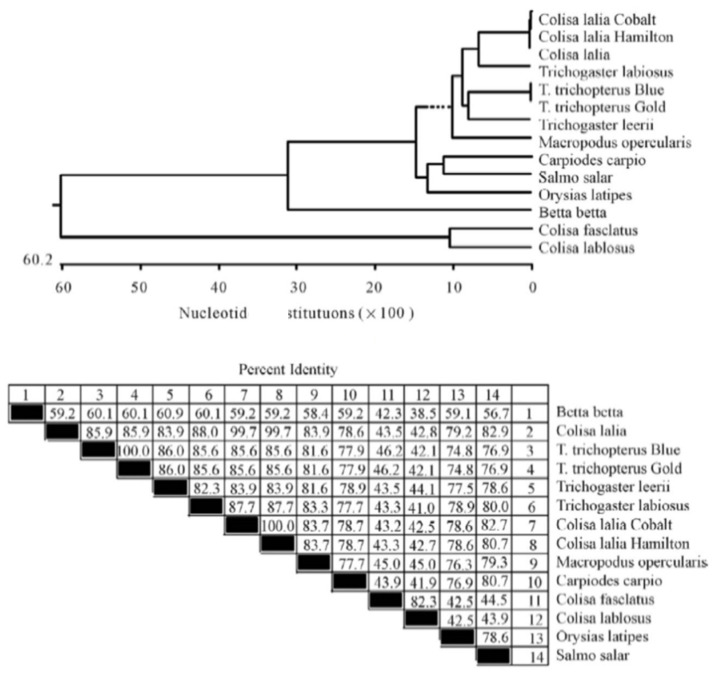
Phylogenetic tree of the *cytochrome b* fragment based on nucleotide sequences of various fish species. The length of each pair of branches represents the distance between sequence pairs, while the units at the bottom of the tree indicate the number of substitution events. The phylogenetic tree was constructed by Clustal W and analysis alignment in the MegAlign program (DNASTAR). Percent DNA sequence identities are also shown [2].

**Figure 9 biology-10-00228-f009:**
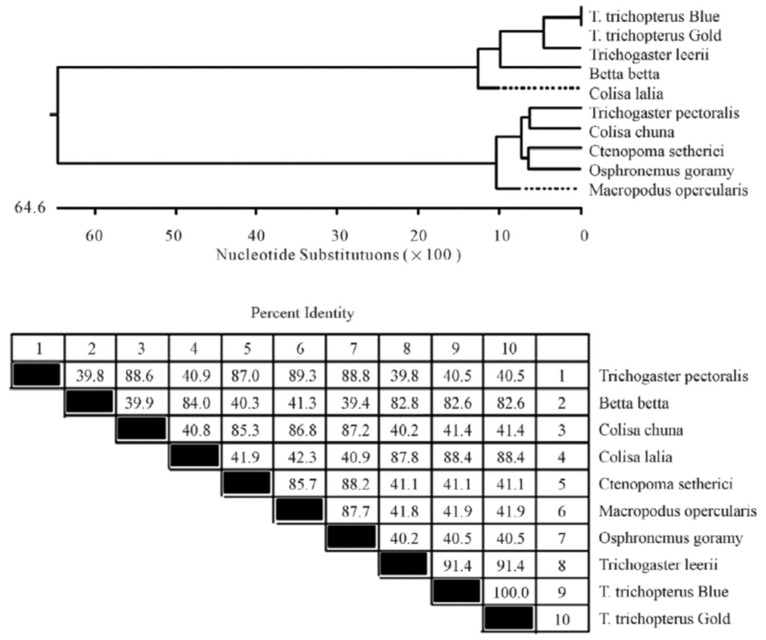
Phylogenetic tree of the *12S rRNA gene* fragment based on nucleotide sequences. The length of each pair of branches represents the distance between sequence pairs, while the units at the bottom of the tree indicate the number of substitution events. The phylogenetic tree was constructed by Clustal W and analysis alignment in the MegAlign program (DNASTAR). Percent DNA sequence identities are also shown [2].

**Figure 10 biology-10-00228-f010:**
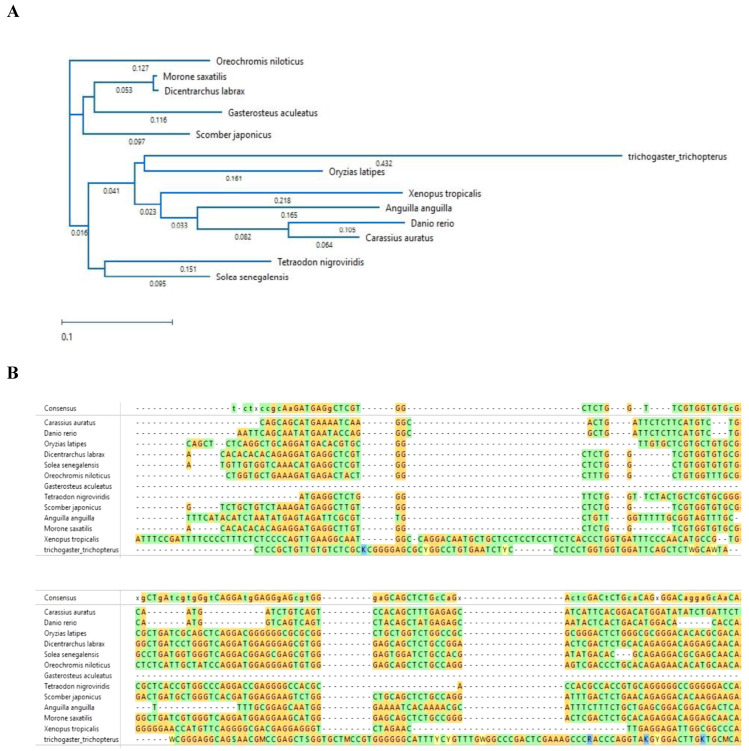
(**A**)**:** Phylogenetic tree showing the relationships between fish *Kiss2* cDNA sequences. The tree was generated using MAFFT alignment in DNASTAR MegAlign PRO. (**B**)**:** Nucleotide and deduced amino-acid sequences [29].

**Figure 11 biology-10-00228-f011:**
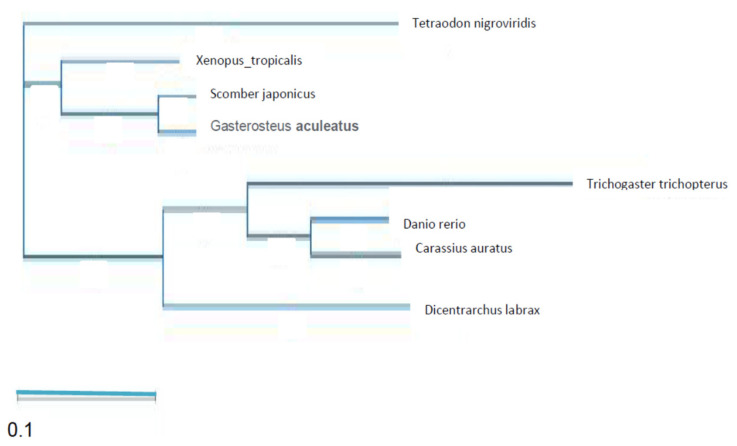
Phylogenetic tree showing the relationship between vertebrate Kiss2r amino acid sequences. The tree was generated by maximum Clustal W alignment in the MegAlign program (DNASTAR). Sequence alignment was conducted using DNA Star MegAlign Pro Clustal Omega. The *Trichogaster trichopterus* sequence was obtained from [29]. All other sequences were obtained from NCBI GenBank.

**Figure 12 biology-10-00228-f012:**
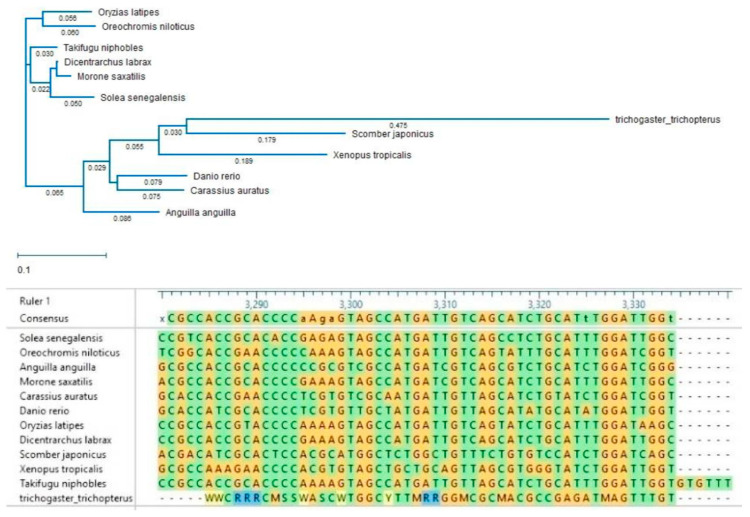
Phylogenetic tree and DNA sequences showing the relationships between vertebrate *Kiss1r* cDNA sequences. The tree was generated using MAFFT alignment in DNASTAR MegAlign PRO [30]. *T. trichopterus* sequence was obtained from [29]. All other sequences were obtained from NCBI GenBank according to the accession numbers provided in [30]. The units of branch length represent nucleotide substitutions per site.

**Figure 13 biology-10-00228-f013:**
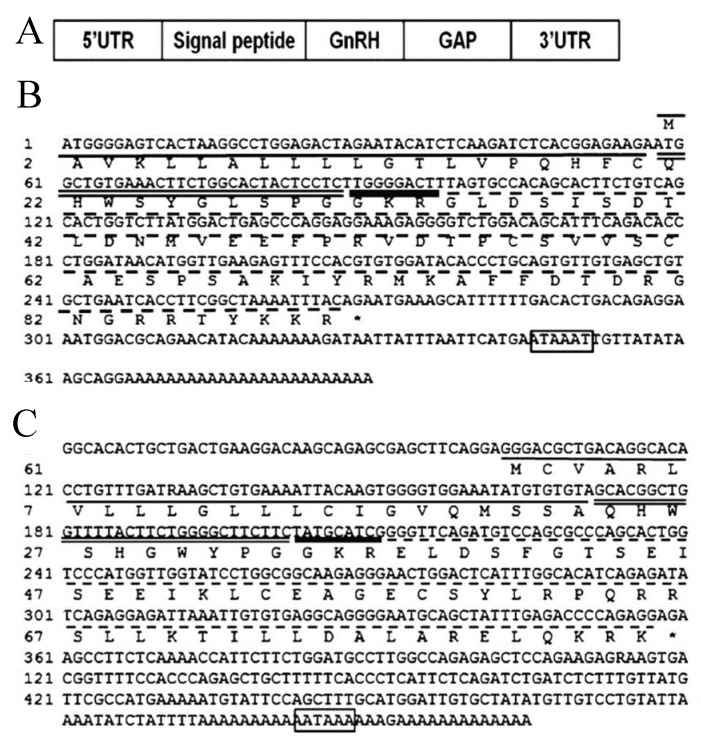
The schematic structure of the prepro GnRH of blue gourami (**A**); cDNA sequences of GnRH1 (**B**) and GnRH2 (**C**). The signal peptide is underlined in black; the GnRH sequence is double-underlined, the cleavage-site position is underlined with a bold black line; the GAP sequence is underlined with a dashed line; and the 3’ poly(a) signal is boxed [31].

**Figure 14 biology-10-00228-f014:**
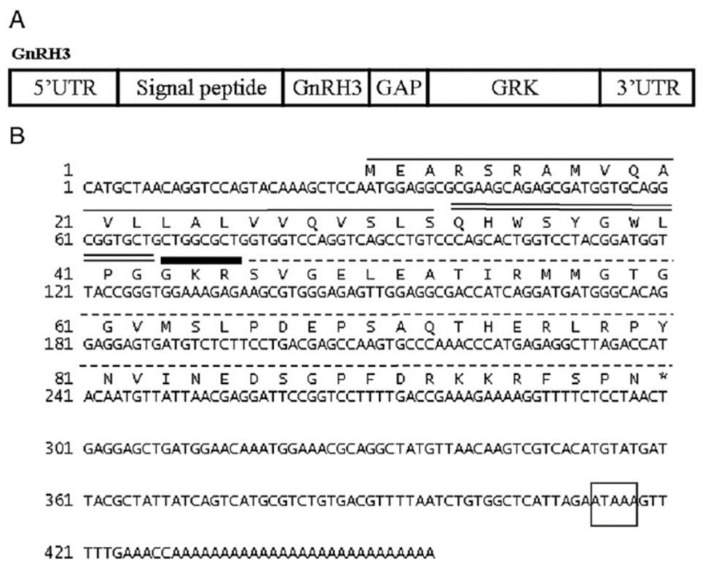
Schematic structure of the Prepro GnRH3 (**A**) and the cDNA and deduced amino acid sequence of GnRH3 (**B**) of the blue gourami. The signal peptide is underlined in black; the GnRH3 sequence is double-underlined and the cleavage-site position is underlined with a bold black line; the GAP sequence is underlined with a dashed black line and the 3′ poly(a) signal is boxed [31].

**Figure 15 biology-10-00228-f015:**
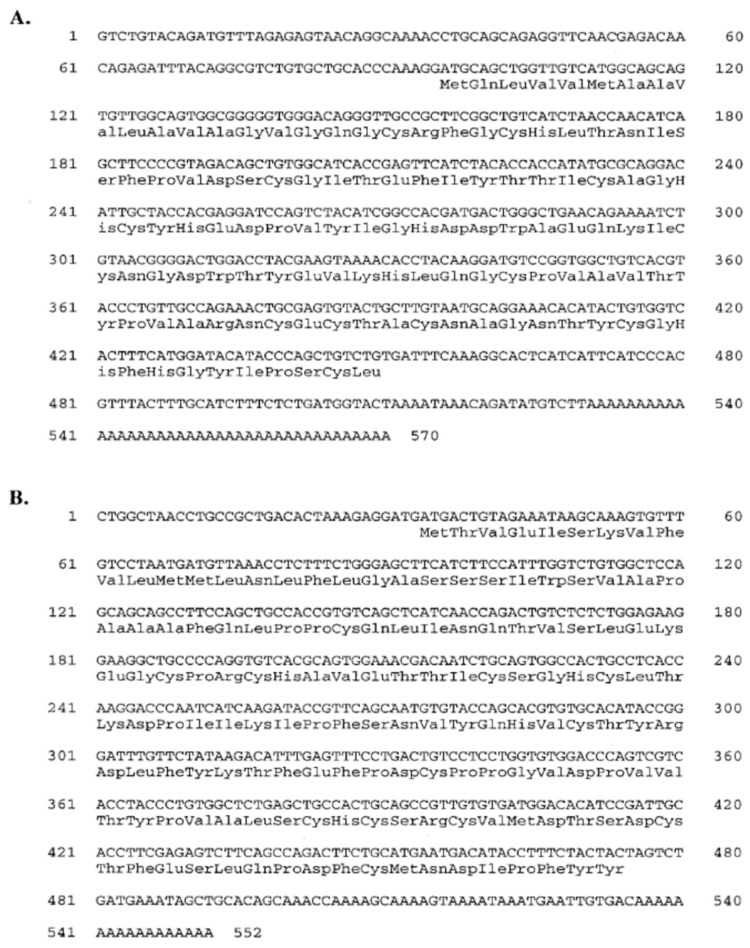
Nucleotide sequences of blue gourami FSH-β (**A**) and LH-β (**B**). The amino acid sequence. of each hormone appears in three-letter code. The GenBank accession numbers of these sequences are AF157630 and AF157631, respectively [34].

**Figure 16 biology-10-00228-f016:**
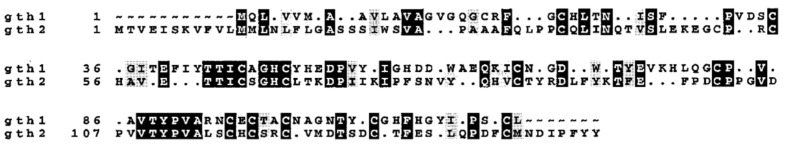
A comparison of the amino acid sequences of FSH-β (gth1) and LH-β (gth2). Identical amino acids are. boxed in black; residues with similar physicochemical properties are boxed in gray [34].

**Figure 17 biology-10-00228-f017:**
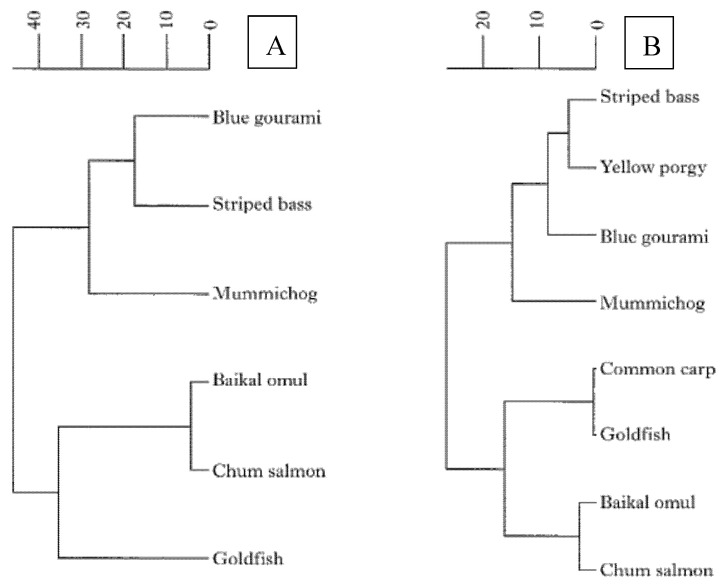
Phylogenetic tree showing the relationships between fish FSH-β (**A**) and LH-β (**B**) sequences. The dendrograms were created by the UPGMA method from similarity matrices that were produced from the corresponding sequence alignments. The scale bar is a measure of the estimated number of amino acid substitutions per 100 residues in pairwise comparisons [34,35].

**Figure 18 biology-10-00228-f018:**
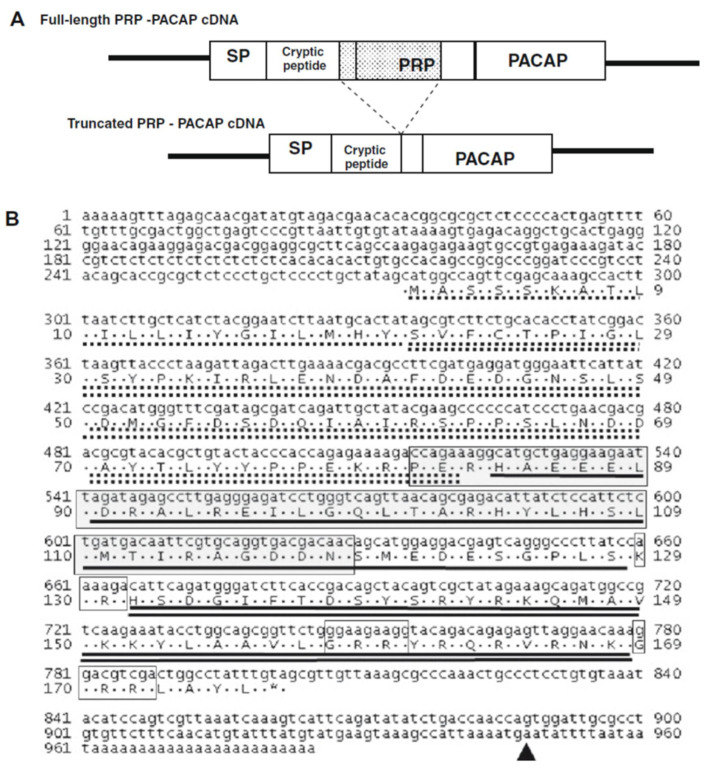
Schematic structures and sequences of PRP–PACAP. (**A**). Schematic structures of the prepro PRP–PACAP cDNA of blue gourami. (**B**). cDNA sequence of PRP–PACAP. The signal peptide is underlined with a dashed black line; the cryptic peptide is underlined with a dashed double line; the PRP sequence is underlined in black; the cleavage-site positions are boxed and the PACAP sequence is double-underlined. The gray box designates the deleted region in the PRP–PACAP short form. The triangle denotes the 3’ poly(A) signal ATTAAA [31,33,36].

**Figure 19 biology-10-00228-f019:**
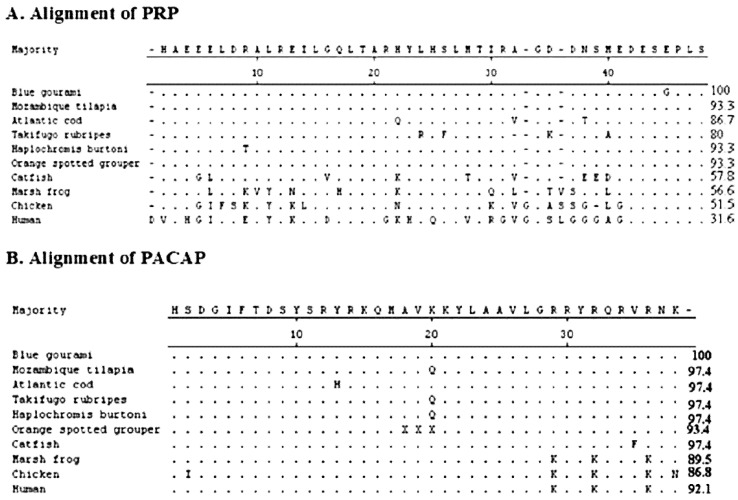
PRP (**A**) and PACAP (**B**) amino acid sequence comparison between blue gourami and other vertebrates. The sequence alignment was conducted using ClustalW and the MegAlign program (DNASTAR) on PRP and PACAP mature peptide sequences. Identical amino acids are represented by dots, and differing residues are shown in the single letter code for amino acids. In contrast to PRP, PACAP is highly conserved [36].

**Figure 20 biology-10-00228-f020:**
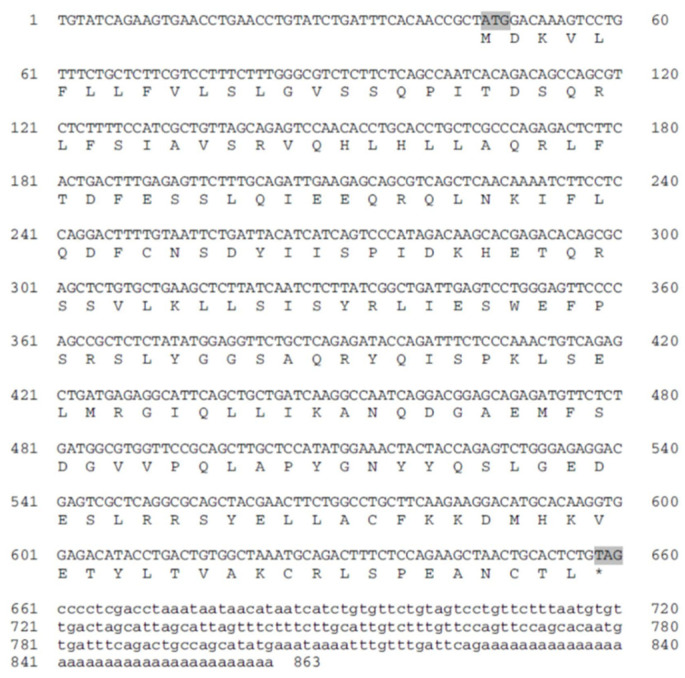
Nucleotide sequence of blue gourami growth hormone (GH) cDNA together with the hormone’s putative amino acid sequence (in three letter code) [42].

**Figure 21 biology-10-00228-f021:**
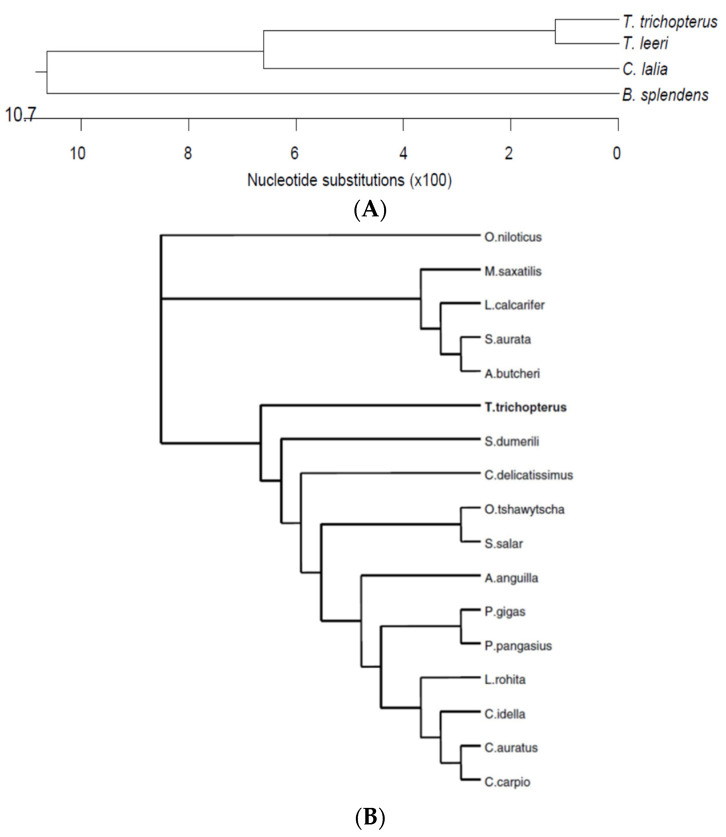
Phylogenetic trees based on the comparison of the deduced amino acid sequence of bgGH to similar Belontiidae family (Anabantoidei suborder) fishes (**A**) and various fish species (**B**) GH found in the GenBank database [41]. Tree was constructed using Neighbor-joining method using Kimura 2 Distance Model, with pairwise deletion and MUSCLE alignment in the MEGA-X software [41].

**Figure 22 biology-10-00228-f022:**
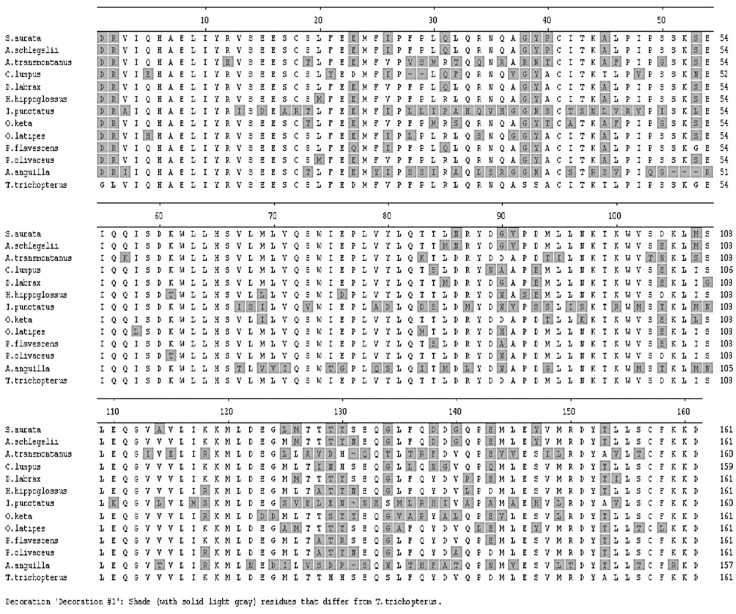
Comparison of cDNA sequences of somatolactin (SL) from blue gourami and various other fish species [44,45]. The phylogenetic tree showing the relationship between SL amino acid sequences compared to those of different teleosts is presented in Figure 23 [45].

**Figure 23 biology-10-00228-f023:**
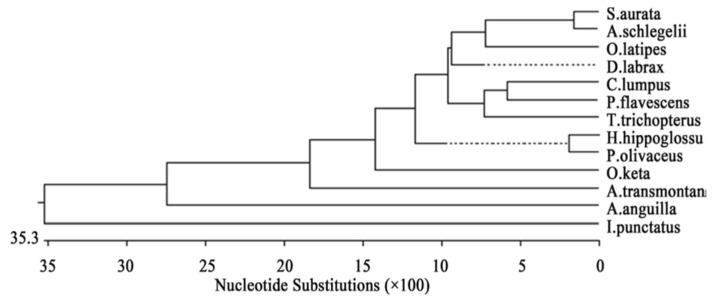
Phylogenetic tree showing the relationship between SL amino acid sequences from blue gourami and other fish species. The tree was generated by maximum Clustal W alignment and the MegAlign program (DNASTAR). All sequences were obtained from NCBI GenBank [44,45]. The highest degree of homology for SL was between *T. trichopterus* and the Perciform *Perca flavescens* and *Cyclopterus lumpus*. The lowest level of homology was observed with *Anguilla anguilla*.

**Figure 24 biology-10-00228-f024:**
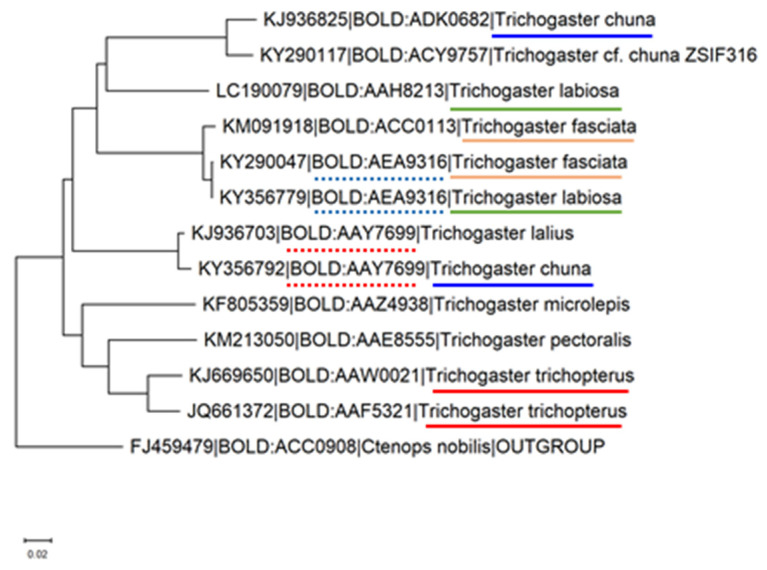
Phylogenetic tree of the COI-5p gene records of the genus Trichogaster. Twelve nucleotide sequences from 8 species were retrieved from BoldSystem [46], such that every species is represented by sequences from unique bins. Tree was constructed using Neighbour-joining method using Kimura 2 Distance Model, with pairwise deletion and MUSCLE alignment in the MEGA-X software [47]. A sequence from the genus Ctenops was used as an outgroup. Leaves are marked with genbank_accession|Bold_bin_id|species. Sequences that belong to the species that appear more than once in the tree are underlined with the same color. Bins that appear more than once are underlined with a dotted line.

**Figure 25 biology-10-00228-f025:**
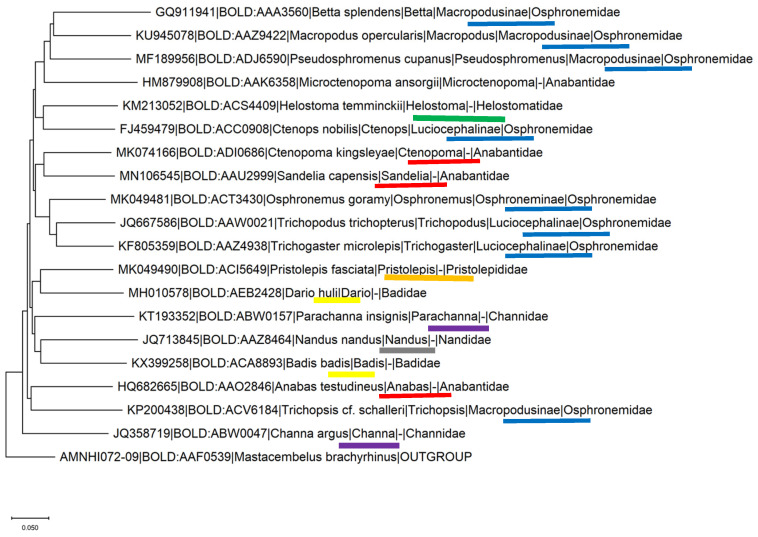
Phylogenetic tree of the COI-5p gene records of the order Anabantiformes. Nineteen nucleotide sequences from different species were retrieved from BoldSystem [46], such that every genus is represented by one sequence. Tree was constructed using Neighbor-joining method using Kimura 2 Distance Model, with pairwise deletion and MUSCLE alignment in the MEGA-X software [47]. A sequence from the species Mastacembelus brachyrhinus that belongs to the order Synbranchiformes was used as an outgroup. Leaves are marked with genbank_accession |Bold_bin_id|species|genus|subfamily|family. Sequences that belong to the same taxonomic family are marked with the same color.

**Table 1 biology-10-00228-t001:** Nucleotide and amino acid similarities (%) between cDNA sequences of GnRH1 of blue gourami and other teleosts [31]. Notes: ^a^ Pejerrey-type GnRH sequence, ^b^ Whitefish-type GnRH sequence.

Order	Species	Accession No.	% Amino Acid Identity	% Nucleotide Identity
**Perciformes (Scombridae)**	*Thunnus thynnus*	EU239500	70.3	74.3
**Perciformes (Serranidae)**	*Epinephelus bruneus*	FJ380047	64.8	72.3
**Perciformes (Cichlidae)**	*Oreochromis niloticus*	AB101665	67	68.2
**Perciformes (Moronidae)**	*Morone chrysops*	DQ000672	61.5	62.6
**Pleuronectiformes**	*Paralichthys olivaceus*	DQ074693	50.5	51.4
	*Verasper variegatus*	HM131600	62.6	70.5
**Atherinomorpha**	*Odontesthes bonariensis* ^a^	AY744689	33	22.6
	*Fundulus heteroclitus*	AB302265	65.9	60.8
	*Coregonus clupeaformis* ^b^		47.3	54.2
**Mugilomorpha**	*Mugil cephalus*	AY373450	62.6	59

**Table 2 biology-10-00228-t002:** Degree of homology between blue gourami (bp) PRL and PRL of other fish classes.

Species	Class/Order	Accession no.	bgPRL (%)
*Perca flavescens*	Perciformes	AY332491	79
*Dientrarchus labrax*		X78723	79
*Spaurus aurata*		AF060541	77
*Paralichthys olivaceus*		AF047616	75
*Onchorhynchus mykiss*	Salmoniformes	M24738	66
*Coregonus autummalis*		Z23114	66
*S. salar*		X84787	66
*Heteropneustes fossilis*	Siluriformes	AF372653	62
*I. punctatus*		AF267990	62
*Hypoththalmichtys molitrix*	Cypriniformes	X61052	62
*Danio rerio*		AY135149	61
*Cyprinus carpio*		X12541	61
*A. japonica*	Anguiliformes	AY158009	59
*A. anguilla*		X69149	59

**Table 3 biology-10-00228-t003:** Degree of nucleotide sequence similarity (%) of blue gourami (*Trichogaster trichopterus*) to different species [2,32,35,36,41,43].

Cytochrome b	Mitochondrial RNA 12S Gene	Growth Hormone	Prolactin	PACAP	GnRH1	GnRH2	GnRH3
*Trichopterus trichopterus* (*gold*) 100%	*Trichogaster trichopterus* (*gold*) 100%	*Lates calcarifer* 84%	*Perca flavescens* 79%	*Oreochromis mossambicus* 94%	*Thunnus thynnus* 74.3%	*Epinephelus bruneus* 78.5%	*Dicentrar chus labrax* 77.4%
*Colisa lalia* 86.6%	*Trichogaster leeri* 91.4%	*Seriola dumerili* 84%	*Dicentrarchus labrax* 79%	*Gadus morhua* 97.4%	*Epinephelu sruneus* 72.3%	*Verasper variegatus* 74.9%	*Pagrus major* 70.2%
*Trichogaster leerii* 86.0%	*Colisa lalia* 88.4%	*Sparus aurata* 83%	*Sparus aurata* 77%	*Takifugu rubripes* 97.4%	*Verasper variegatus* 70.5%	*Paralichthys olivaceus* 73.4%	*Cynoscion nebulosus* 58.1%
*Trichogaster labiosus* 85.6%	*Betta betta* 82.6%	*Acanthopagrus butcheri* 82%	*Paralichthys olivaceus* 77%	*Haplocho misburtoni* 97.4%	*Oreochromis niloticus* 68.2%	*Thunnus thynnus* 72.8%	*Rachycentron canadum* 57.7%
*Macropodus opercularis* 81.6%	*Colisa chuna* 41.4%	*Oreochromis niloticus* 79%	*Onchorhynechus mykiss* 66%	*Epinephelu soioides* 97.4%	*Morone saxatilis* 62.6%	*Morone saxatilis* 72.2%	*Micropog onias undulates* 57.5%
*Cyprinus carpio* 77.9%	*Trichogaster pectoralis* 40.5%	*Morone saxatilis* 79%	*Coregonus autumnalis* 66%		*Fundulus heteroclitus* 60.8%	*Odontesthes bonariensis* 65.2%	*Sciaenops ocellatus* 57.5%
*Betta betta* 58.9%	*Cternopoma setherici* 41.1%	*Caranx delicatissimus* 74%	*Salmo salar* 66%		*Mugil cephalus* 59%	*Mugil cephalus* 65%	
*Salmo salar* 76.9%	*Macropodus opercularis* 41.1%	*Oncorhynchus tshawytscha* 69%	*Heteropneustes fossilis* 62%		*Coregonus clupeaformis* 54.2%	*Fundulus heteroclitus* 61.8%	
*Trichogaster fasciatus* 46.2%	*Osphronemus goramy* 40.5%	*Salmon salar* 68%	*Ictalurus punctatus* 62%		*Odontesthes bonariensis* 22.6%	*Coregonus clupeaformis* 59.6%	
		*Pangasius pangasius* 64%	*Hypophthalmi chthys molitrix* 62%				
		*Pangasinodon gigas* 64%	*Danio rerio* 61%				
		*Cyprinus carpio* 64%	*Cyprinus carpio* 61%				
		*Carassius auratus* 63%	*Anguilla japonica* 59%				

## Data Availability

Not applicable.

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
