# Peer review of "Markers of Genetic Variation in Blue Gourami (Trichogaster trichopterus) as a Model for Labyrinth Fish"

_biology, 2021, doi:10.3390/biology10030228_

Round 1

Reviewer 1 Report

Though the manuscript enclosed some scientific interests, it does not well organized thus hard to follow the meanings especially due to the poor quality of figures and tables. Without grand scale amendment of the manuscript by following indications, I couldn’t recommend this work for the publication.

Major points

  1. Reference problem; For example the Figure 2 is identical to a previously reported article entitled “JBPC  Vol.11 No.4, November 2020. A Qualitative Model of the Interaction of Sexual Behavior and Hormone Gene Transcription in Male Blue Gourami during Reproduction”. Though the author of the aforementioned paper seems to identical to the one of the authors of this manuscript, they must specify the source of the figure by adding adequate reference.
  2. Messy and indiscernible figures and tables; Reading this manuscript was actually more frustrating than I expected, since mostly all images and tables are unintelligible due to poor pixel level. For instance, phylogenetic tree of Figure 11 (A) is not discernable al all.
  3. Fragmented Figure and Tables; For example the Table 2, there is only a fragmented figure in the box! In the case of figure 22, I only recognized a bit of upper fragment.
  4. Disunity of labeling problem; Labeling of the figures must keep unity. The authors had used mixed case word for the labeling of figures. In the case of figure 13, the authors used lowercase while other figures were labeled with capital letter. A bigger problem is that some figures are not properly labeled though the figure legend referred the number of labeling (For example, Figure 5, (A) and (B)). In the main text, I also found several mixed-cases in multisite, e.g., poly(A), Poly(A) etc.
  5. Figure legends are not carefully organized; In figure legend, the authors do not need to rehash contents unmeaningly. e.g., the figure legend of Figure 10. I do not understand why the authors put ‘(Figure 10A,B)’ in its legend.

Reviewer 2 Report

Abstract: “Here, various…..markers” this line need to rephrased.

Introduction: “In their natural habitat…….., and can reach very low concentration, [10].” What amount of low concentration?

In figure 1 New, then went to next paragraph Guinea.

From Introduction, next part 2.Allozyme…. started without any flow. These 2 parts are not connected well. This happened in most of the text. The flow from one part to other is missing.

In second part 2. Allozyme, here full form of MDH was not provided. Second paragraph, first line, “Anabantidae were investigated by starch gel electrophoresis”, came very abrupt. This line looks redundant here. I recommend to check whole manuscript for such sentences.

Part 5 started without bold portion.

Figures and tables are not placed with proper alignment. Please check all of them.

Table 2 was a cut and paste from somewhere. Please make a proper table. If this table was taken from elsewhere please have permission from then and mention from where it was taken.

There were 2 figure 23s.

Table 3 has 5 empty rows.

Round 2

Reviewer 1 Report

Since the authors amended almost all the requirements, the manuscript is now qualified for the publication in “biology”. However, there are still various errors had to be fixed before publication.

1, Figure 18. The triangle seems to be miss-located. PRP– PACAP should be PRP–PACAP.

2, Please check again spacing words. For ex. Figure 19. - Amino acid sequence comparison between blue gourami and other vertebrates of PRP . and PACAP .

  1. Figure 20. The bottom of the figure is not visible.
  2. Please keep unity. For ex. Poly(a) in page 12, poly(A) in page 14.

Author Response

1, Figure 18. The triangle seems to be miss-located. PRP– PACAP should be PRP–PACAP.   The triangle position was corrected 

2, Please check again spacing words. For ex. Figure 19. - Amino acid sequence comparison between blue gourami and other vertebrates of PRP . and PACAP .   The entire text was revised for redundant or unnecessary content, readability, grammar and style, including the indicated location.

• Figure 20. The bottom of the figure is not visible.   Deformation of figures was corrected for better visibility, where possible. The indicated figure is now fully visible.   • Please keep unity. For ex. Poly(a) in page 12, poly(A) in page 14.   This inconsistency has been corrected.

Reviewer 2 Report

I add some comments in the V2 pdf files. Bug the authors need to go thorough the whole manuscript over and over again and work on it, then it will be a good work. This manuscript has good potential, just needs extensive edits. Check line by line for same font, alignment, space, figure pixel, size. etc.

Author Response

I add some comments in the V2 pdf files (attached). Bug the authors need to go thorough the whole manuscript over and over again and work on it, then it will be a good work. This manuscript has good potential, just needs extensive edits. Check line by line for same font, alignment, space, figure pixel, size. etc.   1) The entire text was revised for redundant or unnecessary content, readability, grammar and style.   2) The figure labeling was made more uniform as per a prior request by reviewers.   3) The reference list was checked and corrected.   4) Deformation of figures was corrected for better visibility, where possible.   5) The title and author affiliations were updated and corrected.

Round 3

Reviewer 2 Report

No suggestions.